## [Peer Review File · Nature Communications]

Reviewers' comments:

Reviewer #1 (Remarks to the Author):

The manuscript quantifies the decrease or diminishing of periglacial environments in northern Europe due to projected climate warming. The authors use a statistical approach, where they train the model based on a large number of observations. Not surprisingly, they find that a warmer climate will shift the periglacial realm to higher elevations, but the response is different for different types of periglacial landforms. Is this study necessary when similar statements of course have been given based on climate models? Well, the LSMs related to the GCMs are have a low resolution and are much too coarse (grid cell size = many kilometers depending on model) to reflect the response of periglacial landforms (size = 10s of meters). Besides, the associated LSMs do not include all the processes important for periglacial landform generation. Thus, a statistical approach, allowing for modelling in high spatial resolution, is a valid tool to evaluate the response of complex geomorphological processes to climate change.

The approach is depending on a data set to "train" the statistical model, consisting of the (binary) existence of a landform within a certain areal extent, and explanatory variables related to surficial material, topography and climate (usually temperature or temperature indexes, precipitation and snow cover). Statistical prediction procedures then calculated the probability for a landform to exist based on these variables. The authors use several prediction tools to reduce uncertainties. They finally change the climate variables following different scenarios.

The authors without doubt used a large data set and did a careful analysis of their modelling. The methods in principle are used before, maybe more in ecological studies (distribution and response of e.g. plant species). I guess, novel is the size of the area, and the combination of different prediction models. The principles of including climate variables have been published before by the first author as far as I understand. The methodology is probably more used in ecology than geomorphology, however, within geomorphology the approach could have much more application (e.g. in studies of rapid slope movements and similar), so there might be a wider interest.

Concerning details in the work, it is partly difficult to follow what the authors have done. It is clear that the sort of paper as communication gives limited space to methodological assessments; however, some topics should be clarified, at least in a supplement for some of the comments below.

1. The terms "periglacial" in association to permafrost is confusing in the manuscript. In l. 62 and following permafrost is related to the periglacial features. Here, only the palsa are exclusively associated to permafrost, the other landforms are not by definition. The authors argue also that the absence of deep permafrost makes the area especially vulnerable to climate change. The vulnerability or thermal response to climate change and permafrost is associated to the presence of ground ice and/or organic material, dry bedrock sites (which are common in mountains) react quite fast, too. In line 178 you mention "permafrost flow" in relation to solifluction. I think this is not a correct term, which is mostly related to rock glaciers. Solifluction is not depending on permafrost, and is probably not present in these landforms in northern Scandinavia, at least according several publications from this field.

L. 134 and following permafrost is defined as "relict" in palsa. This is not correct. Relict permafrost is associated to areas where permafrost is found below e.g. taliks, like in southern Siberia, where permafrost is deep and originated through the Pleistocene period, and now overlain by a deep thawed zone. But the permafrost may have formed for several hundred or thousand years ago, however, this is the case for most permafrost.

In l 148 following, you write something about ground ice in slopes, influencing "rapid slope displacement and thaw subsidence". These are topics related to permafrost, but you do not give any information where you have permafrost today in your study site.

2. The LSP sampling is described in other publications. This is ok, however, you may give some more info here. I wonder how the authors selected the 2700 sites, and how they can depict the cryoturbation feature, which obviously need field observations or very good air photos. The latter were available everywhere and cryoturbation or non-sorted circles are good visible? Can it be a bias here, that only big forms are detectable?

3. For the climate data you use a GAM model for estimating temperature to produce the high-resolution grid. For the precipitation you use a (co?) kriging interpolation using topography and proximity to sea. Why not the same procedures for both variables? The downscaling of the climate variables from the climate projections, how did you perform this? And finally, did you do any validation of your results. Obviously, both met stations and rain gauges often are placed in low elevation with subsequent problems with temperature inversions, which are quite frequent at least off the coast. And wrong temperatures would influence also precipitation by snow.

4. You may discuss that a warming of the climate of course will reduce the periglacial realm, defining it based on a lower boundary of e.g. MAAT. However, in terms of landforms, such like palsas, they exist within a relatively close envelope. Palsa would not develop if MAAT becomes too cold because of the lack of organic material production. If you shift the periglacial zone, areas which are now too cold, might become suitable for certain periglacial processes. So, be careful to state that e.g. palsa would disappear. But I agree, the periglacial zone will narrow.

Some minor comments:

l. 62: Reference 30 does not give any information about how deep permafrost is in your study site. The map is quite coarse, and not appropriate for your scale. There have been published products recently in higher resolution

l. 149: This sentence is not justified by your analysis. The same is valid for l.167.

l. 179: The definition of nivation, a bit confusing here. Give a clear definition, which is especially important for a wider audience.

l. 208: What is "rock cover", and in Fig. 4, does "rock cover" play a role for prediction? Reference 24 and 32 are twice.

Fig. 1: The cryoturbation image is difficult to recognize, at least for the wider audience. The same is valid for the nivation. A and B, give graphical scales

Fig. 3: I do not know if I understand the plot, to be honest. You shift the envelope, obviously towards warmer T and more P. Maybe some more explanations are necessary, but this can also be by my lack of understanding here.

Reviewer #2 (Remarks to the Author):

The subject is very interesting and timely with regard to modern climate evolution and its effects. Nowadays most attention is paid to permafrost development and large ice-sheet evolution and their consequences in arctic regions. However, as the authors state, the future evolution of the extensive periglacial environment is highly neglected. The objective of the paper is to fill this gap. The way the authors aim to tackle this objective is by defining a present-day relation between specific periglacial phenomena and climatic and local variables, then to model future climatic conditions and subsequently derive what indicators for periglacial conditions will subsist and thus define the distribution of the future periglacial zone. However, there are some major concerns

about the way how the authors arrive at their present conclusions.

Some major concerns:

1. Since the core of the paper is the evolution of the periglacial realm, the authors have to be clear about the definition of that term. At first they seem to accept the definition of an 'environment dominated by cryogenic (frost-thaw) processes' according to the first reference in their paper. That is fine, but later on they seem to restrict the periglacial environment to the permafrost zone (e.g. line 45) while their study region is limited to permafrost region (line 57-58). The statement in that latter sentence is even false when arguing that 'a broad scale of cryogenic ground processes is coupled with ...the presence of permafrost'. For instance, many cryogenic processes do not need perennial frozen ground (different kinds of patterned ground, slope processes as solifluction, cryoturbation, nivation specifically used in this paper) but only (severe) winter frost.
2. The initial and fundamental step in the study is to establish a relation between current LSP and climatic, topographical and soil properties (lines 68-69, and further). However, this kind of relations appears only in fig. 4 and quite later in the paper without a real discussion (except stating that solifluction and nivation only occur in areas with sufficiently steep slopes, which is really self-evident- lines 104-106). Especially the influence of climate parameters, as freezing and thawing days, on the occurrence of those phenomena require full explanation since they are directly derived from the climate modeling in the authors' scenarios.
3. It is not made clear by the authors why they have chosen a region with continuous to sporadic permafrost (excluding the non-permafrost areas in the periglacial zone) as 3 of the 4 studied periglacial processes (cryoturbation, nivation and solifluction) occur also in a much wider setting of non-permafrost regions, it means also in more southern positions than the study area. It means also that they may occur everywhere in the study region, i.e. they are not limited now by climate but only by topographic and soil conditions. Therefore, one could doubt on the climatic relations derived in fig 4 (freezing-thawing indices): are they not based on too cold conditions at present? Furthermore, it is in the future projections not surprising that exclusively warmer conditions (topography and soils will not change) will decrease the distribution of periglacial phenomena/processes. Thus, the question remains whether the studied phenomena will not persist in future areas with periglacial but non-permafrost conditions.
4. The analysis is essentially based on climate-derived events (periglacial surface processes). That is fine and novel, but therefore this climate-process relations have to be fully explored (my comment 2) and thus I fully agree with the authors' statement in lines 160-161. Otherwise, I have the impression that the present analysis could not learn us much more than the evolution of another climate-derived process, which is the permafrost distribution, in future warming conditions. And such predictions are made already in the literature (but not referenced in the present manuscript).

I realize that these comments will require some additional text, but I have the impression that some space can be found by making the 'experiments of thought' in lines 125-135 and 152-161 somewhat shorter.

Minor and technical comments:

line 45 replace 'are' by 'is' and in line 46 correct typing error in 'lithology'.

Lines 62-67: the authors are not the first to arrive at this conclusion. I suggest they should refer to such model experiments, e.g. in northern Eurasia.

Line 70 : what is meant by 'rock cover': rather fine-grained aeolian sand or loess? Coarse-grained weathered rock rubble? All of them have quite different geomechanical properties.

Ref. List: Some references in the list are incomplete, especially missing journal volume and pages (e.g. references 14 and 17).

Reviewer #3 (Remarks to the Author):

This is an interesting and solid manuscript, in which the authors have developed statistical relationships between different types of periglacial landscapes and the local climate and large-scale

geomorphological context, and then used these relationships to project the changes in the areal extent, elevational distribution, and other features of these landscapes under global warming. In so doing, they borrow the methods of species distribution models to try to calculate the changes to landscape types, which is a novel and interesting application. The results are impressive in that the authors show a clear ability to predict where these landscapes are in the present climate, and dramatic in that they show drastic changes to these landscapes under even small amounts of warming.

My main criticism of this manuscript is that, by (a) focusing on the domain of Europe, which in the current climate is already at the edge of the climate space where one can find periglacial features, and (b) assuming no time lags to be present, the authors have effectively guaranteed the result that they are after, which is a massive reduction in the periglacial domain under warming. How true is this globally? The authors assert that the presence of the Arctic ocean prevents things from migrating northward, which is true, but what is to stop them from contracting eastwards towards the coldest parts of Siberia? In fact, this is exactly what is predicted by climate analogue type approaches (which is effectively what this paper presents): see for example, figure 1a of Koven 2013 (doi: 10.1038/NGEO1801), which shows that this eastward contraction from Scandinavia is the preferred response of climate spatial shifts given the presence of the Arctic Ocean. So one would expect a priori that changes to periglacial realm, which are already marginal in Europe to begin with, would be large. But if the results were expanded to include all of Eurasia, would this still be true?

minor comments:

(1) More detail is needed on the statistical downscaling algorithm used. The results would seem highly sensitive to this, but insufficient detail is given in the section on pages 11-12. E.g., is a baseline climatological correction applied during the downscaling such that the historical climates of the GCMs are effectively specified to be the same?

(2) In figure 4, to what extent are these predictors control variables versus response variables? I'd suspect that the last two (peat and rock cover) are outcomes of the geomorphological processes of interest here rather than drivers, so perhaps it is unfair to use them to predict the occurrence of those features?

(3) In figure 3, how do the actual PDFs of the climate space occurrence, rather than just the boundaries of that space, shift in time?

(4) The authors touch on several implications (carbon budgets, energy budgets, etc) of the changing distributions of periglacial geomorphologic features, but do not attempt to calculate the magnitude of even the sign of what these responses ought to be. Would it be possible to do so?

Reviewer 1

It is clear that the sort of paper as communication gives limited space to methodological assessments; however, some topics should be clarified, at least in a supplement for some of the comments below.

We would like to thank the reviewer for these constructive and helpful comments on the manuscript.

1. The terms “periglacial” in association to permafrost is confusing in the manuscript. In l. 62 and following permafrost is related to the periglacial features. Here, only the palsa are exclusively associated to permafrost, the other landforms are not by definition. The authors argue also that the absence of deep permafrost makes the area especially vulnerable to climate change. The vulnerability or thermal response to climate change and permafrost is associated to the presence of ground ice and/or organic material, dry bedrock sites (which are common in mountains) react quite fast, too.

Response: We have slightly modified the text to highlight that permafrost is not a defining element of the periglacial realm (lines 43-46):

“Although a rapid response is expected⁷, because the broad scale distribution of cryogenic ground processes is coupled with climatic gradients^{17, 18} and often, but not necessarily, with the presence of permafrost^{1, 19}, the details of this and timing are lacking.”

We agree the reviewer that the depth of permafrost is not the only factor defining the climatic sensitivity LSPs and local differences due to e.g. soil and rock type exists. **Therefore we have slightly modified the text in lines 66-70:**

“We argue that the absence of deep permafrost (compared with e.g. High Arctic Canada and Siberia)^{23, 24} means that in general the thermal response of LSP to future climate change is likely to be rapid (perennial ground ice enhances soil-ambient air decoupling²⁵), and thus the region is representative of environments that are especially sensitive to climate change²⁶.”

In line 178 you mention “permafrost flow” in relation to solifluction. I think this is not a correct term, which is mostly related to rock glaciers. Solifluction is not depending on permafrost, and is probably not present in these landforms in northern Scandinavia, at least according several publications from this field.

Response: Following the definition by French (2007) solifluction term encompasses downward movement of topmost soil due to both (1) seasonal frost (c.f. frost creep) and/or (2) “gliding” against permafrost table (c.f. gelifluction). We acknowledge that solifluction is not solely

dependent on the presence of permafrost and that **“permafrost flow” can be confusing term. Thus we have changed it here as “gelifluction (c.f. flow of water-saturated sheet of debris over a permafrost)”**. Unfortunately with our data compiled using visual investigations we are limited to distinguish these sub-processes and instead use the general definition of “solifluction”.

L. 134 and following permafrost is defined as “relict” in palsa. This is not correct. Relict permafrost is associated to areas where permafrost is found below e.g. taliks, like in southern Siberia, where permafrost is deep and originated through the Pleistocene period, and now overlain by a deep thawed zone. But the permafrost may have formed for several hundred or thousand years ago, however, this is the case for most permafrost.

Response: This is a good point and we see that our use of term “relict” was not correct. **Thus we edited this part of the manuscript (lines 147-150):**

“Despite a cover of insulating peat, many of the permanently frozen mires in the region formed during past cold climates are showing evidences of accelerated thawing^{16, 33, 34}.”

In l 148 following, you write something about ground ice in slopes, influencing “rapid slope displacement and thaw subsidence”. These are topics related to permafrost, but you do not give any information where you have permafrost today in your study site.

Response: Ground displacements and thaw subsidence are not limited to permafrost, as in cold-climate regions seasonal ground freezing/thawing can cause substantial damage to e.g. road infrastructure. Our aim was to avoid linking the studied process forms to permafrost as it is evident that permafrost conditions are not needed for these processes to be active (excluding permafrost in palsa mires). **We have slightly modified this sentence to highlight also the role of seasonal ground freezing (lines 164-166):**

“One consequence is that temperature and precipitation driven changes in seasonal and perennial ground ice are likely to increase the risk of rapid slope displacements and thaw subsidence with pronounced societal impact in areas with infrastructure development³⁵.”

Regarding the present permafrost extent of the study area please see the new version of Figure 1 at **page 7** in this revision letter.

2. The LSP sampling is described in other publications. This is ok, however, you may give some more info here. I wonder how the authors selected the 2700 sites, and how they can depict the cryoturbation feature, which obviously need field observations or very good air photos. The latter were available everywhere and cryoturbation or non-sorted circles are good visible? Can it be a bias here, that only big forms are detectable?

Response: Our database of 2917 observation combines targeted field surveys and remotely-sensed observations of the four studied LSPs. For aerial photography interpretation and field surveys the sites could not be randomly chosen due to the extensive size of the study area and vast inaccessible wetlands. Therefore a systematic sampling approach was needed to ensure that key environmental gradients at the study area were fully covered.

For mapping we used aerial photography (spatial resolution of 0.25m²). This is an adequate resolution to detect most of the cryoturbation features which typically exist as a part of larger ensemble, “fields”. Thus the resolution of aerial photography was adequate for clear identification of the activity of these processes. Moreover in our conservative sampling only very evident features were mapped as presence or absence to avoid misinterpretations.

We agree the reviewer that while the very smallest features are detectable only at the field (and thus included to our database), some of these can be missed from the aerial photographs. However, if such small features (less than 50 cm by diameter, e.g. mud-boils) are sporadically present (i.e. not in “group”) then even our unprecedented fine-scale analysis is not able to detect or predict their occurrence.

We have added text to the Methods section to provide more information about the LSP sampling procedure (line 203-212):

“A binary variable (1=presence, 0=absence) of each LSP was established indicating only the evident activity (or absence) of the LSPs. The dataset does not include individually present periglacial micro-features having a diameter less than 50 cm (e.g. mud boils). The process was considered active if even a small area of the process had some indication of activity (absent vegetation cover, mixing of topmost soil, microtopography). The sampling covered the whole study domain and main climatological gradients. A random approach was not feasible, because of the large size of the study area and inaccessible wetlands. To minimize uncertainties related to model extrapolation in time, the data sampling was extended ca. 100 km south of the prediction domain to cover the warmer temperature conditions that will potentially prevail at the prediction domain in the future.”

3. For the climate data you use a GAM model for estimating temperature to produce the high-resolution grid. For the precipitation you use a (co?) kriging interpolation using topography and proximity to sea. Why not the same procedures for both variables? The downscaling of the climate variables from the climate projections, how did you perform this? And finally, did you do any validation of your results. Obviously, both met stations and rain gauges often are placed in low elevation with subsequent problems with temperature inversions, which are quite frequent at least off the coast. And wrong temperatures would influence also precipitation by snow.

Response: This is a good point by the reviewer and we agree that the quality of our climate data is critical to support our findings. Also we agree that the validation results should be available for the readers to improve the transparency of our analyses. Below are responses to each of the raised questions.

Methods for producing temperature and precipitation data:

The air temperature data was produced using generalized additive modelling since in earlier work this method has been proved to be a reasonable method for interpolating climate data. During the revision process of the current manuscript, the paper fully describing the production of the air temperature data used in this study was published:

Aalto J, Riihimäki H, Meineri E, Hylander K, M Luoto. 2017. Revealing topoclimatic heterogeneity using meteorological station data. *International Journal of Climatology*. doi: 10.1002/joc.5020.

This reference is now added to the Methods section (ref 43).

The reason why kriging interpolation (and not GAM) was used for precipitation is that kriging is especially suitable technique for variables with strong spatial autocorrelation. Therefore GAM modelling is not able to capture the often highly localized precipitation patterns e.g. summer rainfall associated with convective processes.

Downscaling of the climate variables:

Here we used a rather simple downscaling scheme where the predicted change by the ensemble average of 23 global climate models were added to our modelled baseline climate (c.f. delta change approach). This adjustment reflecting different greenhouse gas emission scenario and/or time periods was made for each monthly air temperature and precipitation layer. Consequently, the predictor variables (i.e. thawing and freezing degree days, water and snow precipitation) were re-calculated. As stated in the Methods, the results from the statistical downscaling scheme are consistent with climate model projections for high Northern latitudes, which tend to show Arctic amplification of surface and low troposphere temperatures. Please see also response to reviewer #3 comment on the same issue at **page 15** in this revision letter.

Validation of the climate data:

As the validation of air temperature data is now published, we have added only a brief summary of the results to the Methods section (lines 218-220):

“In brief, our modelled monthly average air temperatures agreed well with the observations, with root mean squared error (RMSE) ranging from 0.56 °C to 1.58 °C.”

To show the validation results for monthly average precipitation interpolation we have prepared a new supplementary figure (**Figure S3**).

Figure S3. The agreement between observed and interpolated monthly average precipitation sum (1981-2010) based on a ten-fold random cross-validation, in the terms of adjusted R-squared (R^2) and root mean squared error (RMSE). The red dashed line depicts 1:1 line.

Text have been added to the Methods section (lines 222-224):

“A random 10-fold cross-validation conducted over the gauge data indicated good agreement between measured and interpolated precipitation with RMSE ranging 9.3 -21.7 mm (Fig. S3).”

4. You may discuss that a warming of the climate of course will reduce the periglacial realm, defining it based on a lower boundary of e.g. MAAT. However, in terms of landforms, such like palsas, they exist within a relatively close envelope. Palsa would not develop if MAAT becomes too cold because of the lack of organic material production. If you shift the periglacial zone, areas which are now too cold, might become suitable for certain periglacial processes. So, be careful to state that e.g. palsa would disappear. But I agree, the periglacial zone will narrow.

Response: We agree the reviewer, and thus in line 158 “disappear” has been changed to “to reduce”:

“This means that suitable conditions for LSP at low-elevations are the first to reduce, but potential shifts of LSPs to the highest mountains may be limited by steep topography and rocky or lack of frost-susceptible soil, although nivation processes may continue.”

I. 62: Reference 30 does not give any information about how deep permafrost is in your study site. The map is quite coarse, and not appropriate for you scale. There have been published products recently in higher resolution

Response: We agree the reviewer that this reference alone is too general to support our claim about the absence of deep permafrost. **Therefore we added here a reference (ref 23):**

Westermann *et al.*, 2015. A ground temperature map of the North Atlantic permafrost region based on remote sensing and reanalysis data. *The Cryosphere* 9: 753-790. doi: 10.5194/tcd-9-753-2015

In addition, we have reproduced the Figure 1 to include a panel (B) that shows the modelled permafrost zonation by Westermann et al., 2015 in our study area.

Figure 1. The location of the study domain and LSP observation sites in northernmost Europe. Subfigures (A-B) shows the study area in relation to the circum-Arctic extent of permafrost^{23, 24} indicated as: continuous=90–100 % of the area covered by permafrost, discontinuous=50–90 %, sporadic=10–50 % and isolated= 0-10 %, respectively. Subfigure (C) shows the observations' locations (n=2917) and the relief of the study area. Black rectangles in B and C depict the model prediction domain. Photos show examples of typical surface features of cryogenic land surface processes (scales are only directive): cryoturbation (non-sorted circle) solifluction (solifluction lobes), nivation (snow accumulation sites) and permafrost mounding (palsas), respectively. Photos by J.A. and M.L.

l 149: This sentence is not justified by your analysis. The same is valid for l.167.

Response: Although not directly evident in our analyses, in this sentence we discuss the potential implications of diminishing periglacial activity on geohazards. **This sentence has been slightly modified (lines 164-166):**

“One consequence is that temperature and precipitation driven changes in seasonal and perennial ground ice are likely to increase the risk of rapid slope displacements and thaw subsidence with pronounced societal impact in areas with infrastructure development³⁵.”

and lines 182-185:

“Our analysis, conducted over a wide range of future emission trajectories, indicate that regardless of the climate change mitigation policies the decay of periglacial system is likely to be rapid towards the end of this century.”

l 179: The definition of nivation, a bit confusing here. Give a clear definition, which is especially important for a wider audience.

Response: Here we have modified the definition of nivation to make it clearer for the readers (lines 195-198):

“Nivation is a collective term used to designate all aspects of weathering and fluvial processes, which are intensified and indicated by the presence of local snow accumulation sites³⁶.”

l. 208: What is “rock cover”, and in Fig. 4, does “rock cover” play a role for prediction? Reference 24 and 32 are twice.

Response: We acknowledge that this part of the text has been overly vague. The variable “rock cover” is originally a class in the Corine land cover classification representing exposed non-vegetated rock areas, including e.g. boulder fields and other coarse topsoil material common in

these topographically versatile areas. Thus this variable is linked to the occurrence of LSP through factors such as soil moisture and thermal conductivity. This binary variable (1=rock, 0=no rock) was transformed to a continuous scale (0-1) using a moving 3x3 pixel filter. The figure 4 shows that “rock cover” has very limited effect to the probability of LSP occurrence, at least by generalized boosting method (gbm). For example, we see a slight positive effect for solifluction. Although this could be due to the availability of parental material for solifluction, this effect could also derive from the fact that rock cover is likely to strongly (negatively) correlate with vegetation and other soil properties.

To provide more information about the topographic and soil variables, **we have modified the Methods section of the manuscript (lines 235-247):**

“Three topographic variables were calculated from the digital elevation model (DEM, spatial resolution 50 m x 50 m, data obtained from national land survey institutes of Finland, Sweden and Norway). These included slope angle (representing e.g. mass movement potential), potential annual direct solar radiation⁴⁸ (MJ/cm²/a; surface energy input) and topographic wetness index⁴⁹ (TWI; a proxy for soil moisture). Two soil characteristic variables, rock (i.e. bare rock surfaces including e.g. boulder fields and coarse surficial material) and peat cover, were extracted from a digital land cover classification⁵⁰. Their effect on LSP derive from the thermal-hydrological properties of the soil, as peat has high moisture content and low thermal conductivity (except when frozen) and rock cover is associated with low moisture content and high thermal conductivity¹. The original spatial resolution of the classification is 100 m x 100 m, but the data were processed to a matching resolution of 50 m x 50 m using nearest neighbor interpolation. Binary soil variables were transformed to a continuous scale using a spatial mean of 3 x 3 pixels representing the proportion of a soil variable within a modelling cell¹⁷.”

The duplicate reference has been omitted as suggested.

Fig. 1: The cryoturbation image is difficult to recognize, at least for the wider audience. The same is valid for the nivation. A and B, give graphical scales

Response: This figure has been reproduced to include also graphical scales (please see earlier response).

Fig. 3: I do not know if I understand the plot, to be honest. You shift the envelope, obviously towards warmer T and more P. Maybe some more explanations are necessary, but this can also be my lack of understanding here.

Response: The reviewer has understood correctly; here we contrast the current climate envelope of our study area (this is now clarified in the caption) of the (1) combined and (2) individual LSP occurrences with current and future temperature and precipitation conditions. Thus we are able

to show how the current periglacial realm is shrinking as a function of increasing average temperature and precipitation i.e. the predicted future conditions are likely to be unfavorable for the LSPs. **We have modified the figure caption to provide more information about the meaning of the polygons:**

“Figure 3. The dwindling periglacial climate. The density scatterplots represents the modelled occurrence of the LSP (blue shades; combined spatial extent in the large plot) compared to baseline (climate of 1981-2010) mean annual air temperature and mean annual precipitation in the study area. The black dots indicate individual LSP occurrences. The polygons depicts the convex hulls (i.e. the minimum bounding boxes) of the two climate variables in the study area, and at four time periods and/or climate change scenarios, indicating the shift in climatic conditions in respect to the current periglacial climate realm of the study area.”

Reviewer 2

1. Since the core of the paper is the evolution of the periglacial realm, the authors have to be clear about the definition of that term. At first they seem to accept the definition of an ‘environment dominated by cryogenic (frost-thaw) processes’ according to the first reference in their paper. That is fine, but later on they seem to restrict the periglacial environment to the permafrost zone (e.g. line 45) while their study region is limited to permafrost region (line 57-58). The statement in that latter sentence is even false when arguing that ‘a broad scale of cryogenic ground processes is coupled with ...the presence of permafrost’. For instance, many cryogenic processes do not need perennial frozen ground (different kinds of patterned ground, slope processes as solifluction, cryoturbation, nivation specifically used in this paper) but only (severe) winter frost.

Response: First of all we would like to thank the reviewer for these helpful and constructive comments. This is a very good point and we see the inconsistency in our definition of periglacial realm. We agree the reviewer that LSP are not dependent on the presence of permafrost. However, while not being 1:1 related, commonly the presence of permafrost indicates increased activity of cryogenic ground processes. As we want to stick in our earlier definition of periglacial realm by French (2007, i.e. areas dominated by freeze-thaw processes), we have modified the text:

lines 43-46:

“Although a rapid response is expected⁷, because the broad scale distribution of cryogenic ground processes is coupled with climatic gradients^{17, 18} and often, but not necessarily, with the presence of permafrost^{1,19}, the details of this and timing are lacking.”

While in general our study area is limited to permafrost, we can see from the new version figure 1 that only a small proportion of our study area is associated with continuous or even discontinuous permafrost. Therefore as most of the sporadic and isolated (0-10 % probability of permafrost to occur by Westermann et al., 2015 *The Cryosphere*) permafrost is in *palsa mires*, large proportion of the study area are under influence of seasonal frost processes.

We have added text (lines 56-64) to describe the permafrost conditions of the study area:

“The study area lies between N 68-71° and E 20-26° and covers the transition from continuous to isolated permafrost (Fig. 1), with strong temperature and precipitation gradients (from wet maritime to relatively dry continental) over tens of kilometers. According to a recent modelling estimate²³ continuous and discontinuous permafrost are limited to the highest mountains of the study area (ca. 6 % and 18 % of the observation sites with at least one LSP present, respectively) thus the majority of the region can be classified as sporadic or isolated (50 % and 25 % of the observation sites, respectively) permafrost. This indicates that in large parts of the area LSP are associated with seasonal freeze-thaw processes.”

2. *The initial and fundamental step in the study is to establish a relation between current LSP and climatic, topographical and soil properties (lines 68-69, and further). However, this kind of relations appears only in fig. 4 and quite later in the paper without a real discussion (except stating that solifluction and nivation only occur in areas with sufficiently steep slopes, which is really self-evident- lines 104-106). Especially the influence of climate parameters, as freezing and thawing days, on the occurrence of those phenomena require full explanation since they are directly derived from the climate modeling in the authors' scenarios.*

Response: We agree that these empirical relationships need to be fully described. **Therefore we have edited text in the Results and discussion section (lines 109-114):**

“The probability of active LSP increases towards cold air temperatures (Fig. 4). Permafrost mounding, nivation and solifluction are highly sensitive to TDD (advancing permafrost and snow melt, and soil wetting, respectively), whereas cryoturbation is more constrained by FDD which is strongly linked to frost intensity. In turn, solifluction is controlled by both FDD and TDD affecting frost penetration and spring melt, respectively²⁰.”

and in Methods (lines 227-234):

“TDD and FDD are based on the effective temperature sum of mean daily temperatures above or below 0°C⁴⁵, respectively. Thus they provide an indication of frost intensity and melt period temperatures, being the factors most directly linked to cryogenic LSP^{1, 20}. Water and snow precipitation represent accumulated rainfall (mm) with air temperatures above and below 0°C, respectively. The consideration of snow precipitation is especially important due to its multiple effects on LSP due to insulation properties (constraining the development of permafrost peat mounds and cryoturbation) and snow accumulation (nivation)^{46, 47}.”

3. *It is not made clear by the authors why they have chosen a region with continuous to sporadic permafrost (excluding the non-permafrost areas in the periglacial zone) as 3 of the 4 studied periglacial processes (cryoturbation, nivation and solifluction) occur also in a much wider setting of non-permafrost regions, it means also in more southern positions than the study area.*

Response: We agree the reviewer from the previous version of the manuscript a reader gets the impression that our study domain is entirely underlain by permafrost. We now make clear in the manuscript that the study area covers the transition from continuous to isolated permafrost. The isolated permafrost estimate derives from the modelling efforts by Westermann et al., (2015) depicting 0-10 % chance of permafrost. Saying that, we can expect a large proportion the domain to be under the influence of seasonal freeze-thaw processes. For example from our revised Figure 1, the data by Westermann et al., (2015) we show that continuous/discontinuous permafrost is present only at the highest mountains (6 % and 18 % of the observation sites with at least one LSP present, respectively). In turn sporadic permafrost is likely to occur in the central (i.e. the

continental) parts of the area in peat lands (50 % of the observation sites; e.g. palsa mires). This leads 25 % of our LSP observation to cover areas associated with isolated permafrost (i.e. only a very small chance of permafrost occurrence).

It means also that they may occur everywhere in the study region, i.e. they are not limited now by climate but only by topographic and soil conditions. Therefore, one could doubt on the climatic relations derived in fig 4 (freezing-thawing indices): are they not based on too cold conditions at present? Furthermore, it is in the future projections not surprising that exclusively warmer conditions (topography and soils will not change) will decrease the distribution of periglacial phenomena/processes. Thus, the question remains whether the studied phenomena will not persist in future areas with periglacial but non-permafrost conditions.

In Fig 4 we show that some of the LSPs (permafrost mounding, nivation, solifluction) are especially sensitive to thaw period temperatures. Cryoturbation responses stronger to freezing temperatures compared to thaw temperatures, whereas solifluction is controlled by both TDD and FDD. These empirical findings have physical explanations (please refer to the previous point raised by the reviewer to see text modifications); for example higher thaw period temperatures means that permafrost in mires is likely to melt and snow packs promoting nivation processes are likely to reduce in size. In turn, cryoturbation is mainly driven by freezing temperatures as these are the conditions most directly constraining cryogenic ground processes.

The temperature conditions at our study domain are cold in general, but here we highlight how important it is to consider both freezing and thawing temperatures in LSP modeling. In other words, while the low mean temperatures in the area would allow for LSP activity nearly everywhere, after considering soil and topographical constraints, we show that other factors than low temperatures (i.e. thaw period temperatures) affect LSP activity as well.

As explained in the Methods section, we have expanded our sampling southward to make sure our data cover also warmer conditions, which are likely to exist in our study area in the near future. This will reduce the uncertainty related to extrapolation in time. This is critical as we also show the expected pronounced changes in freeze and thaw period temperatures (as seen in Fig S1).

4. The analysis is essentially based on climate-derived events (periglacial surface processes). That is fine and novel, but therefore this climate-process relations have to be fully explored (my comment 2) and thus I fully agree with the authors' statement in lines 160-161. Otherwise, I have the impression that the present analysis could not learn us much more than the evolution of another climate-derived process, which is the permafrost distribution, in future warming conditions. And such predictions are made already in the literature (but not referenced in the present manuscript).

Response: In the revised version of the manuscript we stress the fact that the LSP distributions are not constrained by permafrost. Please see our earlier responses regarding the inference of the climate-LSP relations and permafrost conditions of the study area.

Minor and technical comments:

line 45 replace 'are' by 'is' and in line 46 correct typing error in 'lithology'.

Response: Edited as suggested.

Lines 62-67: the authors are not the first to arrive at this conclusion. I suggest they should refer to such model experiments, e.g. in northern Eurasia.

Response: We have added new references to this part of the text to support our claims about the sensitivity of our study area to climate change (lines 66-70; the references are listed at the end of this revision letter):

“We argue that the absence of deep permafrost (compared with e.g. High Arctic Canada and Siberia)^{23, 24} means that in general the thermal response of LSP to future climate change is likely to be rapid (perennial ground ice enhances soil-ambient air decoupling²⁵), and thus the region is representative of environments that are especially sensitive to climate change²⁶.”

Line 70 : what is meant by 'rock cover': rather fine-grained aeolian sand or loess? Coarse-grained weathered rock rubble? All of them have quite different geomechanical properties.

Response: Please see our earlier response to the same manner at line 269 in this revision letter (comment by the reviewer #1).

Ref. List: Some references in the list are incomplete, especially missing journal volume and pages (e.g. references 14 and 17).

Response: These have been edited as suggested.

Reviewer 3

My main criticism of this manuscript is that, by (a) focusing on the domain of Europe, which in the current climate is already at the edge of the climate space where one can find periglacial features, and (b) assuming no time lags to be present, the authors have effectively guaranteed the result that they are after, which is a massive reduction in the periglacial domain under warming. How true is this globally? The authors assert that the presence of the Arctic ocean prevent things from migrating northward, which is true, but what is to stop them from contracting Eastwards towards the coldest parts of Siberia? In fact, this is exactly what is predicted by climate analogue type approaches (which is effectively what this paper presents): see for example, figure 1a of Koven 2013 (doi: 10.1038/NGEO1801), which shows that this eastward contraction from Scandinavia is the preferred response of climate spatial shifts given the presence of the Arctic Ocean. So one would expect a priori that changes to periglacial realm, which are already marginal in Europe to begin with, would be large. But if the results were expanded to include all of Eurasia, would this still be true?

Response: First of all we would like to thank the reviewer for the strict and constructive criticism towards the manuscript. We agree the reviewer that the expected changes in periglacial domain are not likely to be globally constant. In turn we recognize this as in our justification for the study area (line 66 ->) where we say that due to absence of deep permafrost (in contrast to e.g. North Eurasia interiors) the thermal response of LSP is likely to be rapid. Thus a response to (a): this location of the study area associated with i) large expected amplitude of climate change, ii) high climatic sensitivity of LSP and iii) highly specialized ecosystems (e.g. palsa mires and nival environments) makes it particularly interesting and important study region for global change impact studies.

If the analyses were expanded to include entire Eurasia, we would expect to see spatially varying response of LSP to climate change. We argue, that in the most interior parts of Eurasia with vast areas of continuous permafrost the response of periglacial processes would be substantially lower (despite rapid climate change) due to thick permafrost that affects the thermal regime of the ground (being more decoupled from ambient atmospheric conditions than area with no permafrost) and enabling the LSPs to remain active longer than e.g. in our study area (c.f. lagged response), where only small parts of the area can be classified as continuous permafrost. In turn, regions with sporadic-discontinuous permafrost (such as western Siberia) the activity seasonal frost-driven LSPs are expected to be severely reduced (similarly than in our study area). Our study area associated with broad gradients in climatic conditions and permafrost (continuous to sporadic/absent) within relatively short geographical distances is a representative of many similar environments across the Earth.

In our study we show that the current periglacial realm of northernmost Europe is not marginal, but rather wide spread (see Fig 5) as roughly 50 % of the study area is covered by at least one of

the studied LSP. This further highlights the role of LSPs being integral part of the functioning of these systems.

Regarding the raised issue of time lags (b), we have discussed this important issue in our previous version of the manuscript (line 145 ->). We agree the reviewer that this is an important issue, but due to adopted empirical modelling approach we are unable to consider (but only discuss based on present case studies as in the text) the future realized lag times, and we are limited to model and predict the suitable environmental conditions for LSP.

We have added the reference Koven (2013) mentioned by the reviewer to line 139 (reference 32) to support our claim about “range expansion” of the periglacial realm.

“The LSP loss will be enhanced in high latitude low-relief continental areas, and in the Northern Hemisphere their projected future climate space is likely to contract dramatically as the Arctic Ocean will effectively limit their "range expansion" northward³².”

Minor comments:

(1) More detail is needed on the statistical downscaling algorithm used. The results would seem highly sensitive to this, but insufficient detail is given in the section on pages 11-12. E.g., is a baseline climatological correction applied during the downscaling such that the historical climates of the GCMs are effectively specified to be the same?

Response: In our study we have used the delta change approach, where we have added the predicted future change by the GCMs to our fine-scale baseline climate of 1981-2010. The method assumes that changes in climates are only relevant at coarse scales and that relationships between variables are maintained towards the future. We acknowledge that approach does not provide a spatially detailed picture of the expected changes, but rather serves as a general representation of the climate change in our study area. Similar method has been used to produce the extremely widely used World Clim data for future conditions by Hijmans et al., (2005, *International Journal of Climatology*), and later used by e.g. Pearson et al., (2013, *Nature Climate Change*). To provide more details about our downscaling scheme, **we have added text to the Methods section of the manuscript (line 253-256):**

“The climate model data depicting the predicted change in mean temperatures and precipitation respect to baseline climate were bilinearly interpolated to a matching resolution of 50 m x 50 m and the predicted change by the GCMs was added to the spatially detailed baseline climate data.”

(2) In figure 4, to what extent are these predictors control variables versus response variables? I'd suspect that the last two (peat and rock cover) are outcomes of the geomorphological processes of interest here rather than drivers, so perhaps it is unfair to use them to predict the occurrence of

those features?

Response: This is an interesting question raised by the reviewer, however we partly disagree with the reasoning. Peat cover: none of the studied cryogenic processes produce peat as an outcome. It is true that processes such as cryoturbation or solifluction mix up the soil horizons, but hardly produce enough organic material as an outcome (at least directly) to form peat lands. Rock cover: we agree that some geomorphological processes e.g. frost weathering in nivation sites produce rock material that cryoturbation consequently sorts and solifluction transports downslope. Most LSPs operate at fine sediment soils (e.g. glacial till) that are susceptible for intense frost action due to the hydrological and thermal properties of the soil. Therefore while we agree that e.g. cryoturbation can increase rock cover through material sorting (at the time scale of hundreds to thousand years), the process is likely to be initiated and active at areas with less rock surfaces (although in our modelling the effect of rock is masked out by the other predictors). This is important since in this study we are investigating the forms showing a present indication of active cryogenic processes.

(3) In figure 3, how to the actual PDFs of the climate space occurrence, rather than just the boundaries of that space, shift in time?

Response: We have prepared a figure to the supplementary material (**Figure S2**), where shift of PDFs of MAT and MAP (constrained by the occurrence of least one LSP) are shown. From the figure below it can be seen how the periglacial climate realm of the study area is shifting towards warmer annual air temperatures and wetter annual precipitation conditions (i.e. unfavorable climate). Therefore this figure highlights the substantial changes that are expected in the periglacial climate realm in our study area.

Figure S2. The kernel density plots (bandwidth = 0.1 and 5 for mean annual air temperature and precipitation, respectively) show the climatic distributions periglacial climate realm of the study area (i.e. combined spatial extent of LSPs) under baseline 1981-2010, 2070-2099 RCP 2.6, 2070-2099 RCP 4.5 and 2070-2099 RCP 8.5 climate conditions. The dashed lines depict the means of the non-smoothed distributions. All distributions differed significantly (t-test, $p \leq 0.001$).

In addition we have added text to the Results and discussion section of the manuscript (line 136-138):

“Therefore these predicted changes in temperature and precipitation regimes will cause the future periglacial climate realm to be shifted to areas that are substantially warmer and wetter than present conditions (Fig. S2).”

(4) The authors touch on several implications (carbon budgets, energy budgets, etc) of the changing distributions of periglacial geomorphologic features, but do not attempt to calculate the magnitude of even the sign of what these responses ought to be. Would it be possible to do so?

Response: This would be an important application and logical next step for investigating the climate feedbacks caused by the changes in cryogenic LSPs. Unfortunately, for current study these questions are far out of the scope and would require a complete re-design of the study, data and analyses. However, our framework of statistical modelling of LSP across broad areas at fine spatial scale can be combined with freely available products of e.g. surface reflectance or soil organic carbon. Therefore, making use of similar spatial modelling scheme than presented in our study could be used to examine the potential landscape scale consequences of shrinking LSP activity on some of the land surface-climate system feedbacks.

Reference list (as in the manuscript)

1. French, H. M. *The periglacial environment* (John Wiley & Sons, Chichester, 2007).
2. Fountain, A. G. *et al.* The Disappearing Cryosphere: Impacts and Ecosystem Responses to Rapid Cryosphere Loss. *Bioscience* **62**, 405-415 (2012).
3. IPCC. in *Climate Change 2013: The Physical Science Basis. Contribution of Working Group I to the Fifth Assessment Report of the Intergovernmental Panel on Climate Change* (ed Stocker, T. F. et al.) 3-29 (Cambridge University Press, Cambridge, United Kingdom and New York, NY, USA, 2013).
4. Straneo, F. & Heimbach, P. North Atlantic warming and the retreat of Greenland's outlet glaciers. *Nature* **504**, 36-43 (2013).
5. Post, E. *et al.* Ecological Dynamics Across the Arctic Associated with Recent Climate Change. *Science* **325**, 1355-1358 (2009).
6. Romanovsky, V. *et al.* Frozen ground. *Global outlook for ice and snow*, 181-200 (2007).
7. Knight, J. & Harrison, S. The impacts of climate change on terrestrial Earth surface systems. *Nature Climate Change* **3**, 24-29 (2013).
8. Schuur, E. A. *et al.* The effect of permafrost thaw on old carbon release and net carbon exchange from tundra. *Nature* **459**, 556-559 (2009).
9. Arneeth, A. *et al.* Terrestrial biogeochemical feedbacks in the climate system. *Nature Geoscience* **3**, 525-532 (2010).
10. Seppälä, M. The origin of palsas. *Geografiska Annaler* **68A**, 141-147 (1986).
11. Washburn, A. L. (1979): *Geocryology*. Edward Arnold, London.
12. Macias-Fauria, M. & Johnson, E. A. Warming-induced upslope advance of subalpine forest is severely limited by geomorphic processes. *Proc. Natl. Acad. Sci. U. S. A.* **110**, 8117-8122 (2013).
13. Frost, G. V., Epstein, H. E., Walker, D. A., Matyshak, G. & Ermokhina, K. Patterned-ground facilitates shrub expansion in Low Arctic tundra. *Environmental Research Letters* **8**, 015035 (2013).
14. Liljedahl, A. K. *et al.* Pan-Arctic ice-wedge degradation in warming permafrost and its influence on tundra hydrology. *Nature Geoscience* **9**, 312-318 (2016).
15. Koven, C. D. *et al.* Permafrost carbon-climate feedbacks accelerate global warming. *Proc. Natl. Acad. Sci. U. S. A.* **108**, 14769-14774 (2011).
16. Christensen, T. R. *et al.* Thawing sub-arctic permafrost: Effects on vegetation and methane emissions. *Geophys. Res. Lett.* **31**, L04501 (2004).
17. Aalto, J. & Luoto, M. Integrating climate and local factors for geomorphological distribution models. *Earth Surf. Process. Landforms* **39**, 1729-1740 (2014).
18. Fronzek, S., Luoto, M. & Carter, T. R. Potential effect of climate change on the distribution of palsa mires in subarctic Fennoscandia. *Climate Research* **32**, 1-12 (2006).
19. Etzelmüller, B. Recent Advances in Mountain Permafrost Research. *Permafrost Periglacial Processes* **24**, 99-107 (2013).
20. Williams, P. J. Climatic factors controlling the distribution of certain frozen ground phenomena. *Geografiska Annaler* **43**, 339-347 (1961).
21. Ridfelt, H., Etzelmüller, B. & Boelhouwers, J. Spatial Analysis of Solifluction Landforms and Process Rates in the Abisko Mountains, Northern Sweden. *Permafrost Periglacial Processes* **21**, 241-255 (2010).
22. le Roux, P. C., Aalto, J. & Luoto, M. Soil moisture's underestimated role in climate change impact modelling in low-energy systems. *Global Change Biol.* **19** (2013).
23. Westermann, S., Østby, T. I., Gislås, K., Schuler, T. V., & Etzelmüller, B. A ground temperature map of the North Atlantic permafrost region based on remote sensing and reanalysis data. *The Cryosphere* **9**, 1303-1319 (2015).
24. Brown, J., Ferrians, O. J. J., Heginbottom, J. A. & Melnikov, E. S. Circum-arctic map of permafrost and ground ice conditions. (2001).
25. Streletskiy, D. A., Sherstiukov, A. B., Frauenfeld, O. W., & Nelson, F. E. Changes in the 1963–2013 shallow ground thermal regime in Russian permafrost regions. *Environmental Research Letters* **10**, 125005 (2015).
26. Guo, D., & Wang, H. CMIP5 permafrost degradation projection: A comparison among different regions. *Journal of Geophysical Research: Atmospheres* **121**, 4499-4517 (2016).
27. Gallien, L., Douzet, R., Pratte, S., Zimmermann, N. E. & Thuiller, W. Invasive species distribution models - how violating the equilibrium assumption can create new insights. *Global Ecol. Biogeogr.* **21**, 1126–1136 (2012).
28. Moss, R. H. *et al.* The next generation of scenarios for climate change research and assessment. *Nature* **463**, 747-756 (2010).
29. Taylor, K. E., Stouffer, R. J. & Meehl, G. A. An overview of CMIP5 and the experiment design. *Bull. Am. Meteorol. Soc.* **93**, 485-498 (2012).
30. Hjort, J. & Luoto, M. Statistical methods for geomorphic distribution modeling. *Treatise on Geomorphology, Shroder J, Jr (ed). Academic Press: San Diego*, 59-73 (2013).
31. Matsuoka, N. Climate and material controls on periglacial soil processes: Toward improving periglacial climate indicators. *Quatern. Res.* **75**, 356-365 (2011).

32. Koven, C. D. Boreal carbon loss due to poleward shift in low-carbon ecosystems. *Nature Geoscience* **6**, 452-456 (2013).
33. Payette, S., Delwaide, A., Caccianiga, M. & Beauchemin, M. Accelerated thawing of subarctic peatland permafrost over the last 50 years. *Geophys. Res. Lett.* **31**, L18208 (2004).
34. Borge, A. F., Westermann, S., Solheim, I., & Etzelmüller, B. Strong degradation of palsas and peat plateaus in northern Norway during the last 60 years. *The Cryosphere Discussions 2016* 1-31 (2016).
35. Liggins, F., Betts, R. A. & McGuire, B. Projected future climate changes in the context of geological and geomorphological hazards. *Philos. Trans. A. Math. Phys. Eng. Sci.* **368**, 2347-2367 (2010).
36. Thorn, C. E. Ground Temperatures and Surficial Transport in Colluvium during Snowpatch Meltout - Colorado Front Range. *Arct. Alp. Res.* **11**, 41-52 (1979).
37. Land Survey of Finland, MapSite. <http://kansalaisen.karttapaikka.fi/kartanhaku/osoitehaku>. (2016).
38. Kartverket, Norge i Bilder. <https://www.norgebilder.no/> (2016).
39. Lantmäteriet, Flygfoton. <http://kartor.eniro.se/>. (2016).
40. Klok, E. & Klein Tank, A. Updated and extended European dataset of daily climate observations. *Int. J. Climatol.* **29**, 1182-1191 (2009).
41. Wood, S. N. Fast stable restricted maximum likelihood and marginal likelihood estimation of semiparametric generalized linear models. *Journal of the Royal Statistical Society: Series B (Statistical Methodology)* **73**, 3-36 (2011).
42. Aalto, J., le Roux, P. C. & Luoto, M. The meso-scale drivers of temperature extremes in high-latitude Fennoscandia. *Clim. Dyn.* **42**, 237-252 (2014).
43. Aalto, J., Riihimäki, H., Meineri, E., Hylander, K., & Luoto, M. Revealing topoclimatic heterogeneity using meteorological station data. *International Journal of Climatology*. In press (2017).
44. Pebesma, E. J. Multivariable geostatistics in S: the gstat package. *Comput. Geosci.* **30**, 683-691 (2004).
45. Carter, T. R., Porter, J. H. & Parry, M. L. Climatic Warming and Crop Potential in Europe - Prospects and Uncertainties. *Global Environmental Change-Human and Policy Dimensions* **1**, 291-312 (1991).
46. Seppälä, M. Snow depth controls palsa growth. *Permafrost and Periglacial Processes* **5**, 283-288 (1994).
47. Edwards, A. C., Scalenghe, R., & Freppaz, M. Changes in the seasonal snow cover of alpine regions and its effect on soil processes: a review. *Quaternary international* **162**, 172-181 (2007).
48. Böhner, J. & Antonić, O. Land-surface parameters specific to topo-climatology. *Dev. Soil Sci.* **33**, 195-226 (2009).
49. Wilson, J. P. Secondary topographic attributes. In 'Terrain analysis: principles and applications'. (Eds JP Wilson, JC Gallant) pp. 87-131 (2000).
50. European Environment Agency, Corine Land Cover 2006 raster data. <http://www.eea.europa.eu/data-and-maps/data/corine-land-cover-2006-raster-1/> (2016).
51. Manabe, S. & Stouffer, R.J. Sensitivity of a global climate model to an increase of CO₂ concentration in the atmosphere. *J. Geophys. Res.* **85**, 5529-5554 (1980).
52. Pithan, F. & Mauritsen, T. Arctic amplification dominated by temperature feedbacks in contemporary climate models. *Nature Geosci.* **7**, 181-184 (2014).
53. McCullagh, P. & Nelder, J. Generalized Linear Models, 2nd edn, Chapman-Hall, London. *Standard book on generalized linear models* (1989).
54. Hastie, T. J. & Tibshirani, R. J. in *Generalized additive models* (CRC Press, 1990).
55. Elith, J., Leathwick, J. R. & Hastie, T. A working guide to boosted regression trees. *J. Anim. Ecol.* **77**, 802-813 (2008).
56. Breiman, L. Random forests. *Machine learning* **45**, 5-32 (2001).
57. Thuiller, W., Georges, D. & Engler, R. biomod2: Ensemble platform for species distribution modeling. R package version 2.1.15. (2013).
58. R Development Core Team. In *R: A language and environment for statistical computing*. R Foundation for Statistical Computing, 2011.
59. Fielding, A. H. & Bell, J. F. A review of methods for the assessment of prediction errors in conservation presence/absence models. *Environ. Conserv.* **24**, 38-49 (1997).
60. Allouche, O., Tsoar, A. & Kadmon, R. Assessing the accuracy of species distribution models: prevalence, kappa and the true skill statistic (TSS). *J. Appl. Ecol.* **43**, 1223-1232 (2006).

Reviewers' comments:

Reviewer #1 (Remarks to the Author):

I mean the revisions raised by the reviewers have mostly been satisfactorily addressed. I have three comments:

1. l. 64. Comment on rapid slope movements

I cannot see how seasonal ground ice (this means ground ice comes during fall/winter and melts during spring) can cause risk of rapid slope displacement, especially in a warmer climate, where maybe the effect of seasonal ground ice is largely reduced. This would imply the opposite. In a permafrost setting, the sentence is justified, this means permanent ice, which has been in the place maybe hundreds or 1000s of years, melts, this gives ground subsidence, and increase the risk for rapid slope movements. The authors should explain this process. I agree that seasonal frost makes damage to infrastructure.

2. l. 88 Comment on sampling

Well, I am not totally convinced here, but I guess it is ok. I am wondering about the absence of the landforms. You obviously go into area where you expect certain landforms to exist. When you then find a place, say with palsa "presence", this means the other landforms get a "absence" for the same location? An then you do this at 2900 places? That I was wondering...

3. l. 197: Comment permafrost map.

It is ok you use the Westermann et al. map, which is based on MODIS data and some re-analysis to fill gaps etc. You may consider using the Nordic permafrost map by Gignas et al., which is published in Permafrost and Periglacial Processes. This map is actually based on interpolated climate data (in principle the same you used), and an equilibrium permafrost model. The map has been validated by Nordic colleagues, and there is a special assessment of palsas included. But this is a matter of "taste". Both maps have their advantages and disadvantages, but concerning palsas the Nordic map is better after my opinion.

Reviewer #2 (Remarks to the Author):

New comments on the revised manuscript:

The authors have made a considerable effort to modify their original manuscript. In that sense, it improved much and especially is more precise and better to understand. My conclusion remains thus the same that this is a most interesting and valuable paper for a wide audience that I advise positively for publication. Nevertheless, I keep a couple of general comments that partially follow my previous comments.

General:

1. I am still puzzled by the used terminology of the periglacial phenomena 'cryoturbation' and 'solifluction'. This might look trivial, but it is not because the terminology really includes the mechanisms involved and the conditions under which they were formed. I see that the authors use quite old references for their definitions and for their derivation of FDD and TDD (e.g. Washburn, Williams), it will be better to use more modern reference as the definitions and conditions given by e.g. French (first reference of the ref. list). Now that the photos illustrating these features are larger (which is a fine improvement) I understand better that what the authors see as 'cryoturbations' are in fact (small-scaled) polygonal patterned soils (not 'non-sorted circles' as

written in the caption of fig. 1). I am not so sure of the authors' claim that the resolution of their air photos had a sufficient resolution for detection as the diameter of these polygons is normally less than 50 cm. Such patterned ground may be caused by different kinds of cryoturbation (differential upheaval according to some theories and load casting- see Vandenberghe 1992), but also by soil cracking due to intense frost. All of these small-scaled patterned grounds do not require permafrost, and the conclusions by the authors that they can form at MAAT up till +2°C and most favorably on (almost) flat terrain is OK. But a more careful handling of the term is strongly suggested. Similarly, the process of 'solifluction' is NOT a process that is by definition occurring in periglacial environments (it is often mentioned for temperate, even tropical environments) and it is NOT formed by frost creep (I 193-195)- I am not quite satisfied with the reply of the authors to my previous comment-, it is a globally occurring slow slope transport mechanism of a soil in plastic state (not water-saturated). Such a plastic state can be caused by locally increased water content due to the formation of an impervious layer resulting from a frozen subsoil. In that case, this specific process is called 'gelifluction' and all phenomena used by the authors in their analysis are in fact gelifluction forms (lobes). They can form in conditions of severe winter frost and do not need necessarily permafrost (as written in I 194, I am not sure of correct citation) - similar as the small-scaled patterned grounds. Palsa's, on the other hand, occur nowadays only in regions with discontinuous or sporadic permafrost and thus by max. MAAT of - 2°C.

2. Fig. 3 contains basic data on which the research is based, especially the 'periglacial climate realm'. Therefore, it is really important to know the source, amount and reliability of the individual LSP occurrences (black dots) as they represent the control data for validation (in addition to the climate data that are well documented now). At least a literature reference is needed.

Details:

L 23: I think it is better to use 'predict' instead of 'suggest'.

L 31: I suggest to insert ', amongst others,' after 'modify'.

Throughout the paper the authors mix present and past tense when describing the operations on which the paper is based. I should prefer one of the two with a preference to the present tense.

L 59: 'modelling': this gives the impression that the concerned permafrost distribution is entirely based on a modelling exercise, but in fact the basic data are field observations.

L 136-138: The inserted sentence, as follow-up from a previous comment, is not very clear. Do you mean that in future the periglacial realm will be shifted to regions that are at present substantially colder? Fig. S2 seems more illustrative than the text to me.

L 442 and 445: replace 'represents' and 'depicts' by 'represent' and 'depict'

Ref.: Vandenberghe, J. 1992 Cryoturbations: a sediment structural analysis. Permafrost and Periglacial Processes 3, 343-352.

Jef Vandenberghe

Amsterdam 5th May 2017

Reviewer #3 (Remarks to the Author):

I think this version of the manuscript is clearer. My major outstanding comment is that I still don't get any sense for how globally representative the result of the diminishing periglacial realm is, and I feel that the authors could have tried to extend their climate envelope approach at a larger scale to do thus; however I accept that they feel that doing so is out of scope of the paper. I therefore support publication in its current form.

Reviewer 1

1. l. 165. Comment on rapid slope movements

I cannot see how seasonal ground ice (this means ground ice comes during fall/winter and melts during spring) can cause risk of rapid slope displacement, especially in a warmer climate, where maybe the effect of seasonal ground ice is largely reduced. This would imply the opposite. In a permafrost setting, the sentence is justified, this means permanent ice, which has been in the place maybe hundreds or 1000s of years, melts, this gives ground subsidence, and increase the risk for rapid slope movements. The authors should explain this process. I agree that seasonal frost makes damage to infrastructure.

Response: We accept the argument from the reviewer that risks for slope displacements, as stated in the earlier version of the manuscript, is related to permafrost rather than seasonal frost.

Therefore we have slightly modified the text in lines 164-167:

“One consequence is that temperature and precipitation driven changes in perennial ground ice are likely to increase the risk of rapid slope displacements and thaw subsidence with pronounced societal impact in areas with infrastructure development.”

2. l. 88 Comment on sampling

Well, I am not totally convinced here, but I guess it is ok. I am wondering about the absence of the landforms. You obviously go into area where you expect certain landforms to exist. When you then find a place, say with palsa “presence”, this means the other landforms get a “absence” for the same location? An then you do this at 2900 places? That I was wondering...

Response: That is correct; if e.g. palsa was present, other three LSPs were assigned as “absence”. We are aware that some of the studied processes partly overlap (e.g. a continuation from cryoturbation at a mountain top to gelifluction at slopes), and such a differentiation can be difficult from aerial images. As a result we have focused only evident process forms and indications of their activity. **We have added text to the Methods section to clarify our sampling protocol (lines 207-210):**

“In a presence of a LSP, others were set as absent, although we are aware that some LSPs can overlap (e.g. a continuation from cryoturbation [mountain top] to gelifluction [slopes]).”

3. l. 197: Comment permafrost map.

It is ok you use the Westermann et al. map, which is based on MODIS data and some re-analysis to fill gaps etc. You may consider using the Nordic permafrost map by Gispnas et al., which is published in Permafrost and Periglacial Processes. This map is actually based on interpolated climate data (in principle the same you used), and an equilibrium permafrost model. The map has been validated by Nordic colleagues, and there is a special assessment of palsas included. But this is a matter of

“taste”. Both maps have their advantages and disadvantages, but concerning palsas the Nordic map is better after my opinion.

Response: Whereas this is a good points by the reviewer, we would prefer to use the Westermann et al., data which provide an estimate of permafrost zonation. Further, this makes the data better comparable to the Brown et al., broad scale map in Fig 1A.

Reviewer 2

1. I am still puzzled by the used terminology of the periglacial phenomena ‘cryoturbation’ and ‘solifluction’. This might look trivial, but it is not because the terminology really includes the mechanisms involved and the conditions under which they were formed. I see that the authors use quite old references for their definitions and for their derivation of FDD and TDD (e.g. Washburn, Williams), it will be better to use more modern reference as the definitions and conditions given by e.g. French (first reference of the ref. list).

Response: We thank Prof. Vandenberghe for these constructive comments. As suggested by the reviewer, we now use more recent references for these definitions (French, 2007 instead of Washburn, 1979).

Now that the photos illustrating these features are larger (which is a fine improvement) I understand better that what the authors see as ‘cryoturbations’ are in fact (small-scaled) polygonal patterned soils (not ‘non-sorted circles’ as written in the caption of fig. 1).

Response: In the figure caption, we have changed “non-sorted circles” to “small-scaled polygonal patterned soils”, as suggested by the reviewer.

I am not so sure of the authors’ claim that the resolution of their air photos had a sufficient resolution for detection as the diameter of these polygons is normally less than 50 cm. Such patterned ground may be caused by different kinds of cryoturbation (differential upheaval according to some theories and load casting- see Vandenberghe 1992), but also by soil cracking due to intense frost. All of these small-scaled patterned grounds do not require permafrost, and the conclusions by the authors that they can form at MAAT up till +2°C and most favorably on (almost) flat terrain is OK. But a more careful handling of the term is strongly suggested.

Response: We agree with the reviewer that cryoturbation includes a variety of sub-processes creating small scale features that can remain undetectable using aerial imagery. This issue of spatial resolution was raised in the previous version of the manuscript where we stated that periglacial micro features (diameter less than 50cm) could not be identified. **We have slightly modified the Methods section (lines 210-211):**

“The dataset does not include individually present periglacial micro-features having a diameter less than 50 cm (e.g. mud boils, soil cracking due to frost action)³⁶.”

We have added a reference Vandenberghe, J. 1992 Cryoturbations: a sediment structural analysis. Permafrost and Periglacial Processes 3, 343-352 (citation 36) to the Methods section to support our definition of cryoturbation and to acknowledge the small-scale cryoturbation features.

*Similarly, the process of ‘solifluction’ is NOT a process that is by definition occurring in periglacial environments (it is often mentioned for temperate, even tropical environments) and it is **NOT formed by frost creep** (l 193-195)- I am not quite satisfied with the reply of the authors to my previous comment-, it is a globally occurring slow slope transport mechanism of a soil in plastic state (not water-saturated). Such a plastic state can be caused by locally increased water content due to the formation of an impervious layer resulting from a frozen subsoil. In that case, this specific process is called ‘**gelifluction**’ and all phenomena used by the authors in their analysis are in fact gelifluction forms (lobes). They can form in conditions of severe winter frost and do not need necessarily permafrost (as written in l 194, I am not sure of correct citation) - similar as the small-scaled patterned grounds. Palsa’s, on the other hand, occur nowadays only in regions with discontinuous or sporadic permafrost and thus by max. MAAT of -2°C.*

Response: We thank Prof. Vandenberghe for this clarification and **thus we have changed the terminology from “solifluction” to “gelifluction” throughout the text. In addition, we have modified Methods section to clarify the definition of gelifluction (and changed the citation from Matsuoka 2011 to French, 2007 [with a direct quote]) (lines 193-200):**

“Gelifluction is a slow mass wasting process caused by high porewater pressure in unconsolidated surface debris where ‘downward percolation of water is limited by frozen ground and where melt of segregated ice lenses provides excess water which reduces internal friction and cohesion in the soil’, creating lobes and terraces¹.”

In addition, all figures have been re-produced in respect to the terminology.

2. Fig. 3 contains basic data on which the research is based, especially the ‘periglacial climate realm’. Therefore, it is really important to know the source, amount and reliability of the individual LSP occurrences (black dots) as they represent the control data for validation (in addition to the climate data that are well documented now). At least a literature reference is needed.

Response: The data in Fig 3 is based on the modelled occurrences of the four LSPs (using empirical data of 2917 observations) and their combined spatial extent (“periglacial realm” in the first panel). The cross-validation statistics of these models are shown in Fig 2, indicating the “reliability” of the modelled occurrences. We think that no literature citation is needed here, since this figure

is based on empirical data and modelling conducted by the authors. **We have added to the figure caption to provide more information of the data that this figure is based on:**

“Figure 3. The dwindling periglacial climate. The density scatterplots represents the modelled occurrence of the LSP (blue shades; combined spatial extent of individual LSPs in the large plot) compared to baseline (climate of 1981-2010) mean annual air temperature and mean annual precipitation in the study area. The black dots indicate individual modelled LSP occurrences, based on empirical data with a total of 2917 observations (see Fig 2 for details and model validation statistics). The polygons depicts the convex hulls (i.e. the minimum bounding boxes) of the two climate variables in the study area, and at four time periods and/or climate change scenarios, indicating the shift in climatic conditions in respect to the current periglacial climate realm of the study area.”

Details:

L 23: I think it is better to use ‘predict’ instead of ‘suggest’.

Response: Edited as suggested.

L 31: I suggest to insert ‘, amongst others,’ after ‘modify’.

Response: Edited as suggested.

Throughout the paper the authors mix present and past tense when describing the operations on which the paper is based. I should prefer one of the two with a preference to the present tense.

Response: We have edited the main text to present tense and methods to past tense, as such seems logical (also advised by the journal editor).

L 59: ‘modelling’: this gives the impression that the concerned permafrost distribution is entirely based on a modelling exercise, but in fact the basic data are field observations.

Response: This sentence has been edited as (line 59 ->):

“According to a recent study²³ continuous and discontinuous permafrost...”

L 136-138: The inserted sentence, as follow-up from a previous comment, is not very clear. Do you mean that in future the periglacial realm will be shifted to regions that are at present substantially colder? Fig. S2 seems more illustrative than the text to me.

Response: We see the potential for confusion here and thus this sentence has been modified as (lines 135-140):

“Therefore these predicted changes in temperature and precipitation regimes will cause the future periglacial realm to reduce markedly in size and the contemporary spatial extent of the periglacial realm will experience a climate that will be substantially warmer and wetter than present conditions (Fig. S2).”

L 442 and 445: replace ‘represents’ and ‘depicts’ by ‘represent’ and ‘depict’

Response: Edited as suggested.